# Systematic Analysis of Literature on the Marketing of Healthcare Systems. Challenges for Russian and Romanian Healthcare Systems

**DOI:** 10.3390/healthcare9060656

**Published:** 2021-05-31

**Authors:** Vladimir Bulatnikov, Cristinel Petrişor Constantin

**Affiliations:** Faculty of Economic Sciences and Business Administration, Transilvania University of Brasov, 500068 Brasov, Romania; vladimir.bulatnikov@unitbv.ro

**Keywords:** healthcare system, health services, healthcare market, healthcare facility, marketing in healthcare, marketing in medicine, perspective directions in healthcare

## Abstract

This paper aims at finding the most dominant ideas about the marketing of healthcare systems highlighted in the mainstream literature, with a focus on Russia and Romania. To reach this goal, a systematic analysis of literature was conducted and various competitive advantages and disadvantages of the medical models that require special attention from the governments are considered. In this respect we examined 106 papers published during 2006 to 2020 found on four scientific databases. They were selected using inclusion and exclusion criteria according to PRISMA methodology. The main findings of the research consist of the opportunity to use marketing tools in order to improve the quality of healthcare systems in the named countries. Thus, using market orientation, the managers of healthcare systems could stimulate the innovation, the efficiency of funds allocation and the quality of medical services. The results will lead to a better quality of population life and to an increasing of life expectancy. As this paper reviews some articles from Russian literature, it can add a new perspective to the topic. These outcomes have implications for government, business environment, and academia, which should cooperate in order to develop the healthcare system using marketing strategies.

## 1. Introduction

In recent times, Russia and Romania have fewer things in common than in the past. Before 1989, the healthcare systems in Russia and Romania were inspired by soviet-type Semashko, funded by the government. After the fall of communism in Romania, a gradual transition to the current system took place and the reform of the healthcare system has been a major project for the government and the Ministry of Health [1]. The change in the direction of politics and economy in Romania in 1989 and in Russia in 1991 led them to developing market oriented economies with their own interests [2] (pp. 13–23).

At the moment, the economic relations between the two countries have a special status due to their ongoing experience in developing small and medium-sized businesses including in the medicine field. Relationships between Russia and Romania have been noticeably improved and intensified at all levels in recent years. A multi-year agreement on the major political directions was concluded between Russia and Romania, which was a complicated process due to several problems arising from the past [3] (pp. 28–50). In this context, the novelty of this paper is in the attempt to combine scientific sources about marketing of healthcare systems from both Romania and Russia. It starts from the assumption that enhancing the quality of life in the general population is an essential component of the country’s economic development. For this reason, the development of a modern healthcare system is a key direction in achieving strategic targets for the country’s economic growth [4] (pp. 87–92). In this respect it is necessary to develop and apply modern methodological tools for evaluating the effectiveness of healthcare activities based on the analysis of their key components. The results of such analyses could offer scientific support for the managerial decisions on the marketing development in this field [5] (pp. 421–431). Marketing management in healthcare consists of a wide range of activities, such as investigation, arranging, execution, and control, which primarily focus on the creation, advancement, and result evaluation in order to fulfill the objectives of a healthcare organization [6] (pp. 20–22). In this regard, it is important to solve the problem of preserving human health and increasing the active growth of individuals’ longevity. Thus, the effectiveness of healthcare systems management in a particular territory is reflected in indicators such as the availability of medical care, the quality of medical services provided, and the achievement of key indicators of industry development [4] (pp. 87–92). The creation and use of innovations in medicine both within the field of avoidance and treatment of illnesses and in the field of socio-economic development of medical institutions is particularly relevant and is one of the mankind priority areas along with energy efficiency and energy saving, nuclear, space and information technologies [7] (p. 232). Starting from these considerations, there is a strong need for theoretical and methodological improvement of assessment procedures and capabilities that can increase the effectiveness of the regional healthcare system.

Taking into account the above mentioned topics, the goal of this research is to identify how the issues about marketing of healthcare systems are highlighted in the mainstream literature, with a focus on Romanian and Russian cases. To reach this goal, 106 articles selected from scientific databases were reviewed. The data collection process followed the PRISMA methodology and aimed to provide answers to two interconnected research questions: Q1. What are the current challenges of healthcare systems in ensuring socio-economic growth?; Q2. How can marketing contribute to the development of healthcare systems? The answers to these questions emphasize that using marketing strategies based on innovative products and processes could be a good solution for a sustainable development of the healthcare systems. Three additional references ([8,9,10]) were used to substantiate the methodological tools.

The article is in four sections. The first section contains an introduction to the field, followed by the research methodology. The research results are presented and discussed in Section 3. Section 4 contains conclusions, proposals and further research directions.

## 2. Research Method

We conducted a systematic literature review to identify the main trends in the marketing of healthcare systems with a focus on Romania and Russia. In practice there are several methods to collate and synthesize evidence for literature reviews, such as PRISMA [8] or Cochrane [9]. Based on the PRISMA methodology, which allows minimizing bad reporting and increasing the transparency in how this systematic review is conducted, and the current paper considers 109 scientific sources that accurately discuss different topics related to healthcare systems. The papers selected for review were published between 2006 and 2020 in around 10 academic journals.

### 2.1. Search Strategy

The selected papers were found in several popular international databases such as PubMed, Scopus, Science Direct and Russian scientific database E-Library. For some articles, which are not open access, the full version was found on social platforms: ResearchGate, Journal of Medicine and Life, Europe PMC, SPB lib, IDEAS, etc. In all databases we used the following terms: “Healthcare Marketing ” OR “Medical Marketing” AND “Healthcare Management”. We chose “Business and Management” as a discipline and “Marketing” as a sub-discipline. Moreover, we collected only papers published from 2006 to 2020 in English or Russian languages. While Romanian authors publish their articles also in English, Russians mainly publish in their native language, but English variants were also found.

### 2.2. Inclusion Criteria

Using the above search criteria in considered databases a huge number of records were found. It counts 5801 articles that correspond to the selected criteria. Using PRISMA methodology, a set of inclusion and exclusion criteria was used (Table 1), where afterwards 834 articles were selected to be potentially suitable for the study.

After we searched through the full text of these articles, another 728 papers were excluded for reasons such as lack of information on the marketing of healthcare systems, or insufficiency of information regarding the European countries for introducing parallel analysis. Articles with lack of information about applied countries or regions in the article were also excluded.

Finally, 106 papers were included in this systematic literature review. In PRISMA Flow Diagram (Figure A1), which was made for a better view of taken steps, the number of included and excluded articles can be found for every step. A year-wise distribution of all selected studies is presented in Figure 1.

The selected studies met the inclusion criteria. The research on the marketing of healthcare systems increased from 2011 to 2013. After a decrease in 2014 the number of articles for every year has again grew up to a peak in 2017, where 14 sources were found on the topic. This evolution highlights the increasing interest of researchers in studying the marketing of healthcare systems.

### 2.3. Risk of Bias in Included Studies

As PRISMA methodology assumes an assessment of the risk of bias in included studies, the Cochrane risk-of-bias tool [10] was used to prevent the issues concerning the findings validity. It includes such key criteria as: random sequence generation; allocation concealment; blinding of participants and personnel; incomplete outcome data; selective outcome reporting; and other sources of bias. Judgments were made by the authors using the following grades: unclear; low risk; and high risk. The overall level of risk was considered moderate as the purpose of this literature review is to highlight a variety of ideas regarding the healthcare marketing and not to collect results of experimental studies that should converge on a common solution.

## 3. Results and Discussions

The research results will be presented and discussed for every research question mentioned above in the introductory section.

### 3.1. Q1. What Are the Current Challenges of Healthcare Systems in Ensuring Socio-Economic Growth?

Healthcare systems have benefited from special attention in social relationships. In the 19th century there were no contractual obligations between doctors and patients, but in the 20th century the liability of the healthcare system for medical practice was introduced and ethical issues have become very important [5] (pp. 421–431). This evolution could be considered a progress toward the approaching of healthcare system from a marketing perspective that “puts the patient at the ”. In this respect, the medical actions are meant to identify and satisf the patient’s needs through a high level of service quality [11] (pp. 440–443).

In any case, in today’s advanced financial conditions, developments in healthcare frameworks are essential and in request. They are a “powerful driving force for development” both within the field of anticipation and treatment of diseases. Their objective is to extend the life anticipation of citizens and fortify wellbeing [12] (pp. 210–214). The “sixth technological order”, developed and spread up to the second half of the current century, is radically changing the nature and structure of society and revealing its fundamental differences from the industrial society that prevailed in the previous centuries [13] (pp. 18–23). This new technological order requires a thorough analysis of today’s problems to identify the dominant economic sector and turn it into a of economic development and in the meantime to bring the economy out of the crisis “like a locomotive” [14] (pp. 200–204). The results of this strategy could lead to a better resource allocation based on quality, rather than price or quantity. The medical services in which should be invested with limited funds have to be identified in order to optimize the healthcare system and reduce the problems that it faces with [15] (pp. 61–67). Additionally, up to the new technological order the world comes to the need to assure a better prepared and people-ed medical staff by implementing competitive academic education programs and knowledge transfer [16] (pp. 1–8). In this respect, the current development of healthcare system in most countries of the world is characterized by a shift in priorities from saving costs for medical care to developing and implementing the most effective ways to allocate resources [17] (p. 5).

#### 3.1.1. The Current Challenges for Healthcare System in Russia

There is a gradual disintegration of the healthcare in Russia that was established in the previous decades. Here the accumulated positive experience, widely recognized and used abroad, is still “rejected without sufficient grounds” and the lack of a comprehensive concept of progressive development prevails [18] (pp. 3–7). Moreover, very modest dynamics of life expectancy at birth is projected, with a very moderate, almost linear growth. A similar picture is shown by the dynamic of the birth rate and mortality among the Russian population [19] (pp. 112–117). Contrary to this, “windows of growth” were found, such as “updating the material and technical base, continuous professional development of personnel, increasing the efficiency of using financial resources, intensifying the use of innovative technologies” [4] (pp. 87–92).

The studies mentioned in the literature were identified many outsider Russian regions with a high differentiation of the subjects in terms of economic efficiency [20] (pp. 66–74). This poor economic development of such regions is highlighted by the regional health index, which is one of the most important components of the “integral indicator of the standard of living” for the population and labor potential [21] (pp. 183–198). The regional health index includes the study of regional socio-economic financial, credit relations and the processes of formation and functioning. It is geographically linked, and therefore the peculiarities of the development of a particular region impose additional conditions [22] (pp. 283–287). Finally, regional targeted programs will be effective only when they are subject to common views, values, attitudes, and ideas prescribed in the socio-economic policy of health development [23] (pp. 157–163).

Nevertheless, additional research is needed to find the reasons for the low efficiency of the national healthcare system and ways to modernize it. Such research could start with a SWOT analyze of the healthcare system in Russia, which is useful for identifying the relationship between the opportunities and threats provided by the environment and the strategic potential of the industry given by its strengths and weaknesses [24] (pp. 24–28). A short SWOT analysis based on the findings in the literature is presented in Table 2.

To sum up, the experts highlight the need to increase “the volume of financing for the industry” in order to assure reasonable quality standards of medical care, especially in rural areas [39] (pp. 28–42). Additionally, there are three main issues to be solved for the current Russian situation: “Reducing alcohol and tobacco consumption, increasing healthcare funding, and efficient spending of industry funds” [40] (p. 592). It should also be noted that healthcare should take into account not only purely economic, but also medical, as well as social aspects of efficiency [41] (pp. 93–105).

#### 3.1.2. The Current Challenges for Healthcare System in Romania

In Romania, the current healthcare insurance system was introduced in 1997. It established a hybrid system controlled both by the Health Insurance Fund and Government, which led to some distortions in resource allocation and even to certain leak of funds out of the medical system. Since its accession, Romania has made efforts to meet the European Union (EU) goals, among which the social protection occupies an important place [42] (pp. 1–7). The Romanian legislation sets health insurance as the main mode of financing the healthcare system, which ensures the access to a set of basic medical services. The health insurance is compulsory to be paid by both employers and employees [43] (pp. 259–267).

In the literature it is highlighted the small number of residents that pay for health insurance in comparison with the number of people that benefit from health services. This discrepancy leads to a poor financing of this system and a need for new viable solutions to increase the budget. [44] (pp. 107–113). In these circumstances, the identification of additional financial resources and the efficient use of existing limited resources should be major concerns for any decision–maker in the Romanian health system [45] (pp. 22–32). Moreover, for innovation, healthcare quality or staff professionalism could be key anchors that potentiate the development the development of healthcare system [11] (pp. 440–443).

A SWOT analysis of the Romanian healthcare system is presented in Table 3, with the aim to identify solutions to avoid the threats and capitalize the opportunities starting from the presumption that “each threat could become an opportunity” [46] (pp. 32–41).

A special attention has to be paid to the strengths and opportunities as well as to the problems that have a negative impact on the national health security.

### 3.2. Q2. How Can Marketing Contribute to the Development of Healthcare Systems?

Marketing may be a “modern, social and administrative prepare by which people and bunches get what they require by making, advertising and trading items with a certain value” [52,53]. Marketing is becoming increasingly popular in medicine, the essence of which is to use commercial sales and marketing methods to solve public health problems [54] (pp. 80–84). Marketing in healthcare is understood as “information processing, including assessment, explanation, modeling, and forecast” of processes in the medical services market, as well as the institution’s own production, sales, and innovation activities [55] (pp. 62–66). This process can be also associated with the functions of “planning, organizing coordination, motivation, control, and financing” and is aimed at achieving the goals set [22] (pp. 283–287). The managers in this sphere must focus on the creation of specialized project selection support systems that provide “comprehensive expertise” and obtain reliable project evaluations. It gives a significant positive social effect at low financial costs for implementation [56] (pp. 108–121). Anyway, in the case of negative demand, the task of marketing is to analyze the reasons for the consumer’s dislike and change their attitude to it, which is called “conversion marketing” [57] (pp. 164–173).

One of step ups in marketing is the development of innovative enterprises that use biotechnology and nanotechnology in order to obtained products with high added value [58] (pp. 112–118). “Medical marketing interactively based on innovation” is the best method for identifying emerging opportunities at a certain time, for stimulating consumption of health services and adopting solutions that can change in time the model of business in medical organizations [53] (pp. 333–335). Reducing the rate of return in old industries encourages entrepreneurs to invest in new products and technologies. During long-term economic crises there is a transition from a strategy of maximizing profits to a strategy of minimizing relative risk. Product innovations mainly appear in the “long-wave depression phase”, and process innovations—in the “recovery phase” [59] (pp. 432–441). However, sometimes, consumers need not so much a new product that can be considered and registered as an invention, but rather to obtain new benefits from an existing product [60] (pp. 70–76). Here the COVID-19 pandemic has had a significant impact. Digitalization of public life has led to the fact that even after the lifting of restrictions, online space plays an important role in human life. “Quarantine was an occasion for companies to start posting and promoting their products on the Internet” [61] (pp. 111–113). In the field of healthcare services, there is a need to realize all the advantages and benefits of digital healthcare, a conscious transition from hierarchical principles of control and regulation to online monitoring based on a community of autonomous network agents is necessary at all levels [62,63,64].

High use of medical technologies with rational use of resources and their accessibility to the general population can significantly increase the opportunities and efficiency of healthcare, positively influence medical and demographic indicators, and improve the image of national healthcare [65] (pp. 22–26). High-tech medical services are using new complex and unique treatment methods, as well as resource-intensive treatment methods with scientifically proven effectiveness, including “cellular technologies, robotic technology, information technologies and genetic engineering” [66] (pp. 4–10) In high-tech assistance, the capabilities of the staff are highly dependent on logistics. Therefore, the development of the internal potential of the health system is “inextricably linked” with the improvement of material and technical support [67] (p. 100).

Currently, the economic development is based on the transition to an innovation-type economy. It needs intensive research and development activities aimed at discovering new technologies [68,69]. Such activities are based on “regional cooperation, coexistence and assistance”, aiming at obtaining coordinated and solidary development of regions of the country” [70] (pp. 172–185). The innovation is also important for healthcare systems taking into account the nation’s objective to develop an integrated information infrastructures [71] (pp. 161–176). The integration of micro-electromechanical systems with microelectronics and wireless interfaces can collect bio-data and help to create artificial intelligence system much faster [72] (pp. 133–138). Here elements of artificial intelligence will find application in medical diagnostics, drugs preparation, radiation therapy, etc. [73] (p. 116).

Innovation itself is considered to pass a full technological cycle from the origin of an idea to its technological development and documentation and finally to the necessary commercial procedures [60] (pp. 70–76). An innovation process in healthcare is a chain of events that transforms an innovation from an idea into a specific product, technology, or service and is used in practice to achieve General goals [66] (pp. 4–10). It involves the unity of the complex ”science—technology—production—distribution” [74] (pp. 54–63). Thus, innovation is the end result of scientific and technical work, provoked by the need for development, which has significant advantages over the previous products [75] (pp. 104–107). If innovation is not in demand for use in the long term, then it is necessary to re-examine whether this was a true innovation at the time of its implementation [76] (pp. 8–11). In this respect, the investments in innovations in the health sector are the riskiest, with a low number of testing alternatives, which increase the overall uncertainty [77] (pp. 3–10). For example, in Russia, the rapid development of innovations in medical sciences, such as “molecular genetics, embryology, and microbiology” often outstrips the public’s readiness to recognize and accept these innovations [78] (p. 14). Therefore, the creation of innovative services must be well managed in order to minimize costs [79] (p. 10), because the resource potential of regions and real opportunities for state support in the formation of projects are often overestimated [80]. However, considering that information flows freely and innovations spread quickly through contacts with patients and suppliers of equipment and drugs this transition could be easy. It can become a business opportunity to enhance the efficiency of healthcare system [53] (pp. 333–335).

In Russia the development of the national high-tech medical care system is faced with the need to solve various problems. Analysis of theoretical developments in recent years shows that insufficient attention is paid to the solution of fundamental issues [81] (pp. 146–152). With a quarter of the world’s natural resources comparable to the leading countries in terms of innovation and high intellectual potential, Russia occupies a modest place in terms of economic development, and significantly lags behind in the competitiveness of its economy [25] (p. 502). There is also an issue of optimizing innovative treatment methods. In each individual institution, similar diseases receive a different amount of medical care [82] (pp. 5–7).

Moreover, the Russian investments in healthcare are not enough for developing innovation [83] (pp. 132–138) and public-private partnerships can have a positive impact by jointly participation of business companies in the modernization process of healthcare system [84,85]. Technology parks can promote innovation by joining “scientific and research institutions, industry facilities, business centers, exhibition venues, educational institutions”, etc. [86] (pp. 159–163). In parallel to technology parks, the economy can be clustered. The main advantage of cluster development is determined by the synergistic effect of joint activities [87] (pp. 1–7). Medical clusters most often arise as associations of competitive organizations based on large medical centers located on the same territory, with a developed scientific base and technology [88] (pp. 167–179). This structure is designed to achieve a synergistic overall effect, including increasing the competitiveness of cluster participants—“first medical aid and healthcare facilities” [89] (pp. 762–774). When creating territorial clusters a synergistic effect is generated, which exceeds the similar effect of vertically and horizontally integrated structures [90] (pp. 219–225). The main task of the medical cluster is to organize interaction and communication between all its actors in the interests of overall healthcare development in the region [91] (pp. 15–17). Using the method of “two-level triad” one of the considered authors systematized the factors contributing to the existence and development of clusters in the medical sector as follows: “innovative capacity, human resources and health infrastructure”, which ensures “intra-cluster connectivity” [92] (pp. 170–175). A medical cluster is a set of institutions united by functional dependency and a single information environment that use innovative scientific developments in their activities. The purpose is to provide competitive emergency and planned medical care [93] (pp. 227–229). Finally, an essential element that ensures the effectiveness of the medical cluster is the participation of various businesses [90] (pp. 219–225). Finding of such companies is the most important of medical clusters [94] (pp. 1–3). The result of the cluster strategy should be the implementation of programs for modernizing healthcare system and improving the quality [95] (pp. 13–17).

There are also side effects to innovative development where the constant increase in healthcare costs associated with the emergence of new medical technologies and medicines is a common trend for all developed countries. However, none of the countries in the world can endlessly increase their spending for these purposes [96] (pp. 80–85). Often, the creators of innovations cannot foresee all the negative consequences. The increasing complexity of knowledge and the transition to interdisciplinary research, which sometimes gives unexpected results, leads to an increase in uncertainty in assessing the possible consequences of using innovations [78] (p. 14).

Thus, improving medical care for the population is possible only if the healthcare system is developed in an innovative way based on the achievements of fundamental and applied science, and new effective medical technologies and drugs are created and introduced into medical practice [4] (pp. 87–92). For this purpose, it is necessary to concentrate financial resources, human resources and medical science resources aimed at solving priority tasks. [97] (pp. 754–762).

Today, health organizations in the marketing environment should identify the multidimensional needs of patients and quickly search, find and direct their resources toward the creation of services to reach these demands. As well as “the steady monitoring of the competition’s situating methodology; the affirmation of the competition’s history, convention and administrative identities; the distinguishing proof of the competition’s showcasing activities in terms of visibility” [11] (pp. 440–443).

To achieve the goals and objectives set for further modernization of healthcare, it is necessary to improve the quality and the effectiveness of the medical services provided to the population [98] (pp. 41–43). Healthcare reforms should be analyzed in the context of broader public relations and the experience of other countries must be considered [99] (pp. 77–80). According to some authors some steps are required: “analysis and revision of the system of cost recovery, the quality control of medical assistance, a change of mentality and cultural level of the population” [100] (pp. 214–221).

A cross-sectoral approach of the healthcare systems management should be implemented because it has to combine constructive efforts of various branches of government and various departments [101] (pp. 61–66). Such policies should achieve a high level of medical service effectiveness [102] (pp. 70–75). The assessment of the effectiveness is based on a chain of three components: “resources-process-result”. Thus, the “resources of the health system”, the “process of providing medical services” and the “result of providing medical services” are considered to be basic synthetic categories of efficiency [103] (pp. 197–213). If a part of the system is not functioning effectively, it has a negative impact not only on the system as a whole, but also on its parts, which can further aggravate the situation [104] (pp. 129–140). There is also a need to improve the regulatory framework by attracting public-private partnerships to the industry, developing registers of high-tech medical care or hospital benchmarking [105] (pp. 146–149).

The development should also consider informal institutions. Taking into account deviations from the self-preservation behavior of residents, non-compliance with labor protection and environmental standards on the part of employers will allow creating such models of economic behavior of actors where saving, rather than consuming, regional health capital is done [106] (pp. 365–378).

Quality is also a proposition of scholars. It is necessary to consider factors such as mass physical education and sports as the determinants of improving the population’s quality of life [107] (pp. 90–92). The economic potential is higher for the organization that uses its capabilities for quality more efficiently. The quality of healthcare services is influenced by the qualification of personnel, the quality physical premises, information and financial resources, etc2] (pp. 95–101). The main areas of work to ensure the quality of medical services are “improvement of the structure, process and result”, which is called the “Donabedian triad” [108] (p. 27). Clearly a medical care framework that has inadequate assets and processes cannot offer quality to the patients [109]. In this context, the healthcare services have to meet customer’s expectations by embracing marketing strategies [11] (pp. 440–443).

### 3.3. Synthesis of Individual Findings

Based on the findings in the literature and the main topics discussed in the analyzed articles, a synthesis of individuals findings was drawn up in Table 4. It is strucured around the two research questions: Q1. What are the current challenges of healthcare systems in ensuring socio-economic growth?; Q2. How can marketing contribute to the development of healthcare systems? Five main topics were identified: Common managerial practices; Evidence of healthcare systems in the analyzed countries; Social relationships; Innovative directions; Implementation of marketing and management strategies.

All articles debate issues regarding the development of healthcare systems but they are mainly focused on specific topics such the ones included in Table 4. The first topic is related to managerial practices, mainly at the macroeconomic level. The second series of articles include studies in the analyzed countries (Russia and Romania) and the third one debate the importance of social relationships in the sustainable development of the healthcare system. The last two topics are mainly focused on the marketing and management strategies and a large part of the articles consider the innovation as the main driver of development.

## 4. Conclusions

The current challenges on healthcare systems in Russia and Romania were considered with pluses and minuses of both systems, which face quite similar problems. Two SWOT analyses were conducted accumulating the main highlighted issues from articles on healthcare of these two countries. The need for assuring better functionality of these systems by considering both economic and social aspects is generally revealed in the literature. The adoption of marketing philosophy and the implementation of long-term market-oriented strategies could be the best solutions for solving the open healthcare issues. In this respect, the innovative development is considered the best solution for applying the marketing strategies. Innovation is the best method for identifying emerging opportunities at a certain time, for stimulating consumption of health services and adopting solutions. The use of innovative products and processes in healthcare systems can lead to large increments in openings and productivity. In the scholars’ opinions, Russia has taken significant steps in adoption of innovations in medicine, mainly in the fields of “molecular genetics, embryology, and microbiology”. Such best practices could be also adopted by Romania in order to solve various problems that the healthcare system faces with.

As regards the authors’ propositions for improving the healthcare systems, the experience of other countries was offered: cross-sectoral approach of the health policy management; improvement of the regulatory framework; developing registers of high-tech medical care; hospital benchmarking; quality management, etc.

The result of this literature review article may give experiences for directing future investigations regarding the marketing of healthcare systems in Russia and Romania. Thus, the findings of this article could help government, business and academia to concentrate their efforts and policies on the development of the current healthcare system, especially using marketing tools.

Directions for improving the healthcare systems should mainly focus on reducing diseases, disabilities and mortality among the population; improving the quality and accessibility of healthcare; developing a competitive market of innovative medical services. The creation of an effective healthcare system could lead to improving the quality of the population life and increasing life expectancy.

### Limitations and Future Works

There are three minor limitations in the article that must be regarded in the future research. First, the number ideas identified in the analyzed studies is quite low and they are not enough for combining Russian and Romanian experience in marketing of healthcare systems. Secondly, the low number of up-to-date studies within the field of marketing of healthcare systems limited the analysis and thirdly, there is a lack of experience in the healthcare systems marketing in the analyzed countries.

At this very moment we presented a set of characteristics and steps for designing the marketing of healthcare systems of Russia and Romania found in the literature. Future research will be concentrated on finding the opinions of experts and mass-population regarding the best solutions that marketing could offer for improving the quality of healthcare systems in Russia and Romania.

## Figures and Tables

**Figure 1 healthcare-09-00656-f001:**
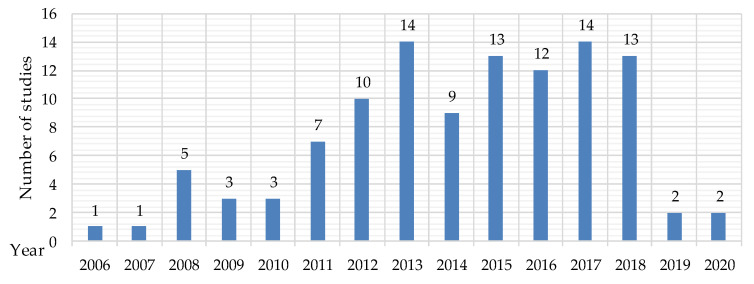
Year-wise distribution of selected studies.

**Table 1 healthcare-09-00656-t001:** Inclusion and exclusion criteria.

Inclusion Criteria	Exclusion Criteria
It must have evaluation of at least one future result on implementing marketing strategy, or detailed in conclusion;	Papers that claims the term “marketing in healthcare” but in reality they are about someone’s particular business;
2. Paper has to be released during 2006 and 2020; 3. Article should contain information about European country or region, prioritized on Romania and Russia.	2. Reviews and editorials; 3. Articles with lack of information on applying country or region; 4. Non-scientific articles; 5. Articles with high risk of bias.

**Table 2 healthcare-09-00656-t002:** SWOT analysis of healthcare system in Russia.

**Strengths**	**Weaknesses**
“In recent years, the growth rate of average life expectancy in Russia is one of the highest in the world” [25] (p. 502); Overall, maternal, child and infant mortality are reduced [26] (pp. 300–305); Constant decrease in alcohol consumption per capita and the number of drug addicts [27] (pp. 108–116); Implementing the project “Electronic healthcare” [28] (pp. 113–116); “The material and technical base of healthcare has been reorganized and improved” [29] (pp. 159–162).	“Inefficiency of the obligatory medical insurance system” [30] Reduction of Federal budget expenditures on healthcare both in absolute and percentage terms [31]; The share of public spending on health as a percentage of GDP is low [32]; High depreciation of fixed assets (57% at the beginning of 2017) [32].
**Threats**	**Opportunities**
Increase in mass spread of diseases, increase in cases of injuries and poisoning [33] (p. 53); Increased availability of psychoactive and psychotropic substances for illegal consumption [31]; “Shortcomings of the state policy in the field of health protection” [28] (pp. 113–116); Increasing the share of the state healthcare expenditures as a total amount of expenditures and as a % of GDP [34] (pp. 139–143); Increase of health workers salary [31]; “Low quality training of health workers and the quality of medical services provided” [35].	“The salaries of health workers will be increased” [36] (pp. 277–278); Increasing share of government spending on health in total spending [37]; “Increasing interest of business representatives and state authorities in the development of medical infrastructure” [38]; Developing an information system for monitoring the medicines traceability in consumption in order to eliminate falsifications and counterfeit [30].

**Table 3 healthcare-09-00656-t003:** SWOT analysis of healthcare system in Romania.

**Strengths**	**Weaknesses**
” Macroeconomic stability” [1]; “The new law on healthcare reform” [46] (pp. 32–41); Relatively predictable and favorable fiscal policy [1]; Centers of excellence that assure medical care for people regardless of the area they live in [46] (pp. 32–41).	“Lack of an effective marketing” [47] (pp. 472–476); “Corruption practices and ineffective judicial system” [1]; Poor implementation of healthcare policy [48] (pp. 1–13); Lack of institutions autonomy [46] (pp. 32–41); “Poverty of the population” [47] (pp. 472–476); “Demographic crisis and aging population” [1]; “Poorly developed telecommunication networks” [49].
**Opportunities**	**Threats**
Opportunities for investments that come from other EU countries [50]; Recoil of socio-economic determinants [46] (pp. 32–41); “Adoption of the single Euro currency” [47] (pp. 472–476); European tourism in healthcare [1]; Health promotion [46] (pp. 32–41).	“Poor environmental conditions” [46] (pp. 32–41); “European economic crisis and slow down” [1]; “Unreasonable healthcare behavior related to health risk factors” [46] (pp. 32–41); “Migration of the highly skilled labor” [47] (pp. 472); “Shortages of well qualified specialists” [1]; “High incidence of contagious and chronic diseases” [51].

**Table 4 healthcare-09-00656-t004:** Synthesis of individual findings.

Research Question	Topics
Q1. What are the current challenges of healthcare systems in ensuring socio-economic growth?	Common managerial practices: [1,2,3,4,5,6,11,12,13,14,15,16,17]; Evidence of healthcare systems in the analyzed countries: [18,19,20,21,22,23,24,25,26,27,28,29,30,31,32,33,34,35,36,37,38,39,40,41,42,43,44,45,46,47,48,49,50,51]; Social relationships: [1,26,27,28,29,30,31,32,33,34,35,36,37,38,46,47,48,49,50,51].
Q2. How can marketing contribute to the development of healthcare systems?	Innovative directions: [7,29,58,59,60,61,62,63,64,65,66,67,68,69,70,71,72,73,74,75,76,77,78,79,80,81,82,83,84,85,86]; Implementation of marketing and management strategies: [25,52,53,54,55,56,57,83,84,85,86,87,88,89,90,91,92,93,94,95,96,97,98,99,100,101,102,103,104,105,106,107,108,109].

## Data Availability

Not applicable.

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
