# Peer review of "Systematic Analysis of Literature on the Marketing of Healthcare Systems. Challenges for Russian and Romanian Healthcare Systems"

_healthcare, 2021, doi:10.3390/healthcare9060656_

Round 1

Reviewer 1 Report

Dear Authors .

Thank you for the opportunity to revise the paper.

The topics covered in the article are interesting and within the scope of Healthcare.

However, I have a few reservations and suggestions for improving the paper:

  1. the main objection is the poor quality of the archangel language. I recommend proofreading here.
  2. introduction - suggests adding brief information on the organization of the text
  3. Methodology - PRISMA assumes the use of scientific databases such as Web of Science or PubMed or Scopus. Reserachgate is not a scientific database, but only a social medium for scientists. Drawing (otherwise interesting results) from non-recommended sources may undermine the scientific value of the paper.
  4. I propose to distinguish sections : search strategy
  5. The number of articles selected for analysis is very large. Did the authors consider introducing further selection criteria? I suggest you choose only one language instead of two.
  6. PRISMA assumes an assessment of the risk of bias. Please respond to this in your paper.

Higgins JPT, Altman DG. Chapter 8: Assessing risk of bias in included studies. In: Higgins JPT, Green S, eds. Cochrane handbook for systematic reviews of interventions version 5.0.0 [updated February 2008]. The Cochrane Collaboration, 2008.

  1. Analysis of PRISMa- checklist- it is necessary to synthesise the individual findings. I propose to refer to the individual articles included in the analysis. This is usually done in a table. Only then is a synthesis of the results presented
  2. The choice of hypotheses is surprising. PRISMA usually uses a research question because it is a systematic analysis of the literature
  3. The title should include" systematic analysis of the literature"

I recommend reading PRISMA's methodology on the website:

http://www.prisma-statement.org/

I believe that the article requires major revision

Kind Regards

Reviewer 2 Report

The paper has been written well by securing the readability. It includes an array of information on healthcare systems both in Russia and Romania. However, I have the following concerns on the current version of paper:

  1. The study provides a systematic literature review on marketing of healthcare systems in Russia and Romania. The review is aimed at answering 05 central research questions. However, the topic highlights only on challenges faced by healthcare systems in both countries. Thus, the topic does not reflect what the paper basically talks about.
  2. Also, systematic reviews generally suffer from “publication bias” which follows from the selective publication of papers depending on the magnitude or direction of the results. The paper does not include anything on how such biases are detected and rectified if any.
  3. Do all 145 papers discuss about all 05 questions? Otherwise, some research questions may be addressed on the basis of a few papers which again leads to biases of results. It would be better, if you could choose one or two central research questions and review the literature around them. An array of research questions for one systematic review may not be appropriate.

Reviewer 3 Report

The article is based on solid literature research on the management aspects of two different health systems. There is no originality in the research, being based on the work of other authors, but it has a very practical and almost didactic function. It is a well structured article, even if not easy to read. The research methodology is well explained, as are the inclusion and exclusion criteria of the articles examined. The discussion is complete and detailed. It can be accepted in the present form.

Round 2

Reviewer 1 Report

Dear Authors
Thank you for addressing the suggested corrections and making all necessary amendments and improvements to the text. The article now meets the methodological requirements and is ready for publication 

Congratulations!